# BUILDING THE BUILDING BLOCKS: FROM SIMPLIFICATION TO WINNING TREES IN GENETIC PROGRAMMING

## ABSTRACT

Genetic Programming (GP) represents a powerful paradigm in diverse real-world applications. While GP can reach optimal (or at least "good-enough") solutions for many problems, such solutions are not without deficiencies. A frequent issue stems from the representation perspective where GP evolves solutions that contain unnecessary parts, known as program bloat.

This paper first investigates a combination of deterministic and random simplification to simplify the solutions while having a (relatively) small influence on the solution fitness. Afterward, we use the solutions to extract their subtrees, which we denote as winning trees. The winning trees can be used to initialize the population for the new GP run and result in improved convergence and fitness, provided some conditions on the size of solutions and winning trees are fulfilled. To experimentally validate our approach, we consider several synthetic benchmark problems and real-world symbolic regression problems.

## 1 INTRODUCTION

Genetic programming (GP) is an evolutionary algorithm (EA) used to automatically generate computer programs to solve specific tasks (Koza, 1992). Up to now, GP has been used in diverse domains like image analysis (Varniab et al., 2020), cybersecurity (Picek et al., 2018), or scheduling (Nguyen et al., 2017). What differentiates GP from other types of evolutionary algorithms is the fact that the solutions are commonly represented as expression trees. Naturally, due to a specific solution representation, appropriate variation operators need to be used. Unfortunately, while being very successful in diverse domains, GP also faces certain representation-specific problems. Indeed, the GP process will introduce redundancy and functionally useless sections of programs, commonly denoted as program bloat (Koza, 1992; Blickle et al., 1994).

Program bloat can cause various issues to the GP process. As a consequence of bloat, the evolution process can prematurely terminate and explore large parts of the search space that are not promising. Since the solutions with program bloat will be more complex than needed, the interpretability of solutions will suffer. Additionally, program bloat can be connected with some forms of building blocks, i.e., parts of the solutions that can be combined to form even more fit solutions.

As there are no universal ways to deal with program bloat, this problem is difficult. Commonly used methods are mostly based on applying various forms of simplification to reduce the complexity of a solution and eliminate any useless details. While such techniques can reduce program bloat, there are further questions to consider. For instance, simplification can be done at the end of the evolution process (Hooper & Flann, 1996) but also during the evolution process (Wong & Zhang, 2006; 2007). As simplification will reduce redundancy, it can also have negative effects on the diversity of the obtained solutions and decrease the protection of useful building blocks within solutions from the destructive nature of the crossover operator (Blickle et al., 1994).

Numerous works have explored the issues of program bloat or how to simplify the GP solutions as discussed in Section 2. What is more, multiple works have discussed the notion of building blocks for GP (Langdon & Poli, 2002), but a widely accepted definition for building blocks is still not agreed upon. A somewhat similar concept to building blocks has been recently discussed in the deep learning community. There, the Lottery Ticket Hypothesis for artificial neural networks discussed

how "...a randomly initialized dense neural network contains a sub-network that is initialized such that, when trained in isolation, can match the test accuracy of the original network after training for at most the same number of iterations.", which surmises the existence of a "winning ticket" for neural networks (Frankle & Carbin, 2019). This opens up the question of the existence of something like winning tickets for GP.

This work considers simplifications (both exact and heuristic ones) in tree-based GP applied for symbolic regression, and we show that building blocks play an important role. More precisely, our contributions are:

- We present a new technique to conduct heuristic simplification for GP. We combine the heuristic simplification with the exact one, allowing us to reduce the tree sizes significantly.
- We discuss a new concept in GP that we denote as winning trees. While there are similarities between winning trees and building blocks, the main differences stem from the fact that the winning trees have constraints on fitness and tree depth. We introduce the notion of winning trees to help us understand better how to guide the evolutionary process to highly-fit solutions.
- We provide extensive experimental analysis to showcase that the evolutionary process when using winning trees can result in 1) more fit solutions, 2) faster convergence, and 3) a more robust evolutionary process. Additionally, we discuss the depth constraints for winning trees and show they work well only if of sufficient size, coupled with the appropriate solution depth.

## 2 RELATED WORK

Several studies have already dealt with the problem of simplifying the expression trees evolved by GP. These methods can roughly be divided into two groups, methods that simplify the expression without changing the behavior of the original expression and those which, by simplifying the expression, change its behavior compared to the original one. The former group consists mostly of methods that perform algebraic simplification. These methods apply predefined algebraic rules to replace a part of an expression with a simpler but equivalent one. In the literature, such methods are usually denoted as algebraic simplification or editing. Such a method was applied by (Wong & Zhang, 2006), and (Zhang et al., 2006) online during the evolution; the authors defined 30 algebraic rules which were applied to the evolved trees in the population. The algebraic simplification method was combined with hashing, which helped to determine equivalent expressions (Wong & Zhang, 2006; 2007). These methods hash sub-expressions and determine whether certain expressions are equivalent or not based on the hash value. This helps the algebraic procedure to detect situations that would otherwise go undetected.

The second group of simplification methods introduces modifications that can change the expression's behavior. The idea is to detect parts of the expression that are not meaningful and remove them, although this might affect the numerical outcome. The main question is how to determine which parts of the expression should be eliminated. (Kinzett et al., 2008; 2010) proposed the application of a numerical simplification based on node contribution to reduce the complexity of the evolved trees. (Song et al., 2009) pruned trees by removing a child of an operator with two arguments if, when removed, the removal does not change the output of the node beyond a given threshold. (Garcia-Almanza & Tsang, 2006) proposed a method for tree pruning applied to decision trees used to generate classification rules for financial stock markets. The method consists of extracting rules from a decision tree, evaluating them, and then pruning the rules that did not achieve the expected pruning threshold in the evaluation phase. A constant sub-tree pruning method was proposed by (Rockett, 2020). This pruning scheme replaces a subtree with a constant value. The constant is calculated as the expectation of the subtree, which is replaced on the entire dataset.

Building blocks were first discussed in the context of genetic algorithms as short, low-order, and highly fit schemata that are sampled, recombined, and resampled to provide solutions of higher fitness (Goldberg, 1989). Intuitively, building blocks are small parts of a solution that can be formed into larger, more fit components through genetic operators. Defining a building block for GP is more difficult, where some common (traditional) interpretations are a subtree to a solution tree (Altenberg, 1994), a rooted subtree (Rosca, 1997), and a block of code (Angeline & Kinnear, 1996). In this work, we will follow the work from Langdon and Poli that define building blocks as parts of the solution

that can be combined to form more fit solutions but without any constraints on fitness, length, or order of building blocks (Langdon & Poli, 2002). We note a recent work by O'Neill et al. where the authors discuss transfer learning for GP and how to create valuable material as building blocks dynamically (O'Neill et al., 2017). Still, we consider this work orthogonal to ours as we do not aim at transferability while imposing other constraints.

Since its definition, several studies which focus on the lottery ticket hypothesis have been made. For instance, (Chen et al., 2020) used observations from the work on the lottery ticket hypothesis (Frankle & Carbin, 2019) to find trainable and transferable subnetworks in pre-trained BERT models, which are commonly used in natural language processing. (Frankle et al., 2020) used iterative magnitude pruning (IMP) to determine "whether a neural network optimizes to the same, linearly connected minimum under different samples of SGD noise".

## 3 METHODOLOGY

To evaluate the winning trees hypothesis for GP, we need to consider the expressions that GP obtains. There is no guarantee that the expressions obtained by GP are exact (i.e., they represent the exact solution), and at the same time, they can contain many unnecessary subexpressions. This is often true when GP is executed for longer periods, and bloat starts occurring (Luke & Panait, 2006). To mitigate this problem, we introduce two methods for reducing the complexity of the expression trees. We denote these methods as *simplification*, for the exact procedure and *pruning*, for the heuristic procedure.

### 3.1 SIMPLIFICATION

The goal of the simplification approach is to simplify the expression without changing its behavior. Therefore, this approach replaces expressions with equivalent but simpler ones. One such example would be to replace the expression $x + 0$ with $x$ since the first one includes two redundant symbols. The rules for simplification in our experiments are based on common algebraic axioms and aim to reduce the size of the expression, such as the example above, and the complete list is given in Appendix A, Table 7. The applied rules are not exhaustive, and more rules can be defined and used. However, the given set of rules has shown to be enough to simplify the obtained expressions.

The simplification is performed using the Compare-Match algorithm proposed in (Steyaert & Flajolet, 1983). The algorithm includes two recursive functions, *Match* and *Compare*. The *Match* function iterates through the expression tree from the root node through all of its subtrees and determines whether it can find a pattern $P$ in the given expression $T$. If a match is found, then $P$ is replaced by a simpler expression defined for that pattern. Finding patterns is performed by the *Compare* function, which traverses through the expression tree and returns *true* if it could locate pattern $P$ in the expression tree, otherwise it returns *false*. The *Compare* function applies an additional *Map* function to retrieve the information about the located pattern. The outline of the method is given in Appendix A, Algorithm 1.

### 3.2 PRUNING

Unlike simplification, which is exact and does not change the behavior of the expression, the pruning method allows the expression to be simplified by removing parts of it, which will directly influence its output. This means that the expression does not necessarily need to have the same output value for a given input before and after pruning. This technique is performed so that certain elements in the expression are simply replaced by a neutral element for a given operator. For example, in the expression $+XY$, where $X$ and $Y$ represent arbitrary subexpressions, the expression can be pruned by removing either operand and replacing it with a neutral element for summation, which is 0. Thus, it is possible to obtain either $+X0$ or $+0Y$. In that way, one part of the expression can be removed while keeping the rest of the expression fixed. Each binary operator has its neutral element, which for the summation and subtraction are 0, whereas, for multiplication and division, they are equal to 1. Unary operators do not have a neutral element as they represent a single expression. Thus, they are not considered in the pruning process.

The removal of certain subexpressions can change the behavior of the entire expression. To observe how the modifications influence the expression's behavior (the semantics), we monitor the change in fitness between the original expression and the pruned one. If the fitness improves or stays the same, the change is accepted. Our approach also allows pruning to be performed even if it reduces the fitness compared to the original expression; however, a limit is imposed on how much the simplified expression can be worse than the original one.

The procedure that performs the pruning starts from the root node of the expression tree and traverses the tree. At each node, it replaces the current node with a corresponding neutral element. Then, the new expression is evaluated, and its fitness is compared to the fitness of the original expression. The new expression is accepted if it does not degrade the performance of the original expression beyond a certain degree. In our experiments, we set that the solution should not be degraded beyond 15% of its original quality. Naturally, this parameter can be arbitrarily chosen to balance the complexity and quality of the expressions. If the change is not accepted, then the procedure continues the traversal through the expression tree. Otherwise, if the change was accepted, the procedure is restarted at the root node and traverses the tree again to check whether it is possible to perform additional simplifications. The procedure stops when both the entire expression tree is traversed, and no modification is performed. To the best of our knowledge, this kind of pruning approach for GP has not been applied before.

### 3.3 Search for Winning Trees

We define winning trees for GP as parts (subtrees) of a solution (tree) that can be formed into larger, more fit components through the use of genetic operators. Differing from building blocks, winning trees have:

- constraints on the fitness value - as they are formed as subtrees of highly fit solutions (trees), and
- constraints on the size – as they cannot be arbitrarily small or large.

Note that winning trees are actually subtrees, but we denote them as trees for simplicity.

The candidates for the winning trees will be obtained by first identifying highly fit simplified expressions in the following procedure. A full GP evolution will be executed $n$ times, and the best individual from each run on the training set will be stored. When $n$ individuals are obtained, each of them is simplified by the previously outlined methods in the following way. First, the exact simplification is performed, followed by the heuristic pruning, and then again the exact simplification. The goal of the first exact simplification is to reduce the complexity of the original expression if possible. Since the heuristic pruning needs to evaluate the expression after each change, it is beneficial to start with an already simplified expression to reduce the number of possible evaluations (especially in cases where the fitness function is more complex). After pruning, the exact simplification is again invoked to remove possible redundant elements introduced by the heuristic simplification (for example, when introducing neutral elements in the expression).

After the process mentioned above, the $n$ obtained expressions are considered simplified. Therefore, in our approach, the simplification is performed offline after the GP run has finished. The simplified expressions can then be used as the basis to create subtrees that will be used for the initialization of the initial population of a new GP run. However, using only the complete expressions would not make sense as it would mean that GP would be starting from the best solutions and trying to improve them further (emulating a setting with a longer evolution process). Therefore, smaller subexpressions (winning trees) that are part of the original ones are inserted into the initial population. This is performed by selecting random subtrees (with a defined depth) from the $n$ available expressions. These selected subtrees are then used as individuals in the starting population of a new GP run.

A depth limit for the selected expressions is additionally imposed. If the GP is allowed to evolve expression trees of depth $d$, then the maximum depth of the randomly selected subtrees is equal to $\frac{d}{2}$. In that way, we wish to include smaller subtrees (relative to the maximal depth of the tree) but also ensure that not only small subtrees are included in the new population. Note that we limit the minimal tree depth to 2 for all settings. What is more, the limit on $\frac{d}{2}$ represents an upper bound, which means we also use smaller winning trees. The value $\frac{d}{2}$ was selected arbitrarily, and we leave further investigation on various depths for winning trees for future work.

## 4 EXPERIMENTAL SETUP

### 4.1 DATASETS

To test our hypothesis, we use two symbolic regression benchmark sets. One set is based on existing synthetic benchmark functions whose exact target expressions are known and, as such, can be fitted without error. The second benchmark set consists of several real-world regression datasets for which the exact solution is not known.

Table 8 in Appendix A shows a selected set of ten synthetic symbolic regression problems (Oliveira et al., 2018). This set was selected to include functions of different forms and complexities. All selected functions are either one or two-dimensional. A training set consisting of 10 000 instances and a test set of 1 000 instances were generated for all functions. Both sets contain instances sampled uniformly from the interval $[-5, 5]$. Table 9 in Appendix A denotes the properties of the datasets for the five real-world regression problems (Dua & Graff, 2017). These sets were divided into the training set containing 70% of instances and the test set containing 30% of instances. Compared to the synthetic benchmark problems, these problems have fewer instances that can be used for training and contain more input variables.

### 4.2 GENETIC PROGRAMMING

The applied GP uses the steady-state tournament selection with a tournament consisting of three individuals. The population size of the algorithm was set to 500 individuals and the mutation probability to 0.3. The tree depths of 6, 8, and 10 will be used in experiments to test the hypothesis with different expression sizes. The crossover operators included the subtree, context preserving, one point, size fair, and uniform operators, whereas the set of mutation operators contains the subtree, shrink, permutation, node replacement, node complement, hoist, and Gauss mutation operators (Poli et al., 2008). Each time individuals are crossed over or mutated, the operators that will be applied are randomly selected from the set of crossover and mutation operators. The function set contains the addition, subtraction, multiplication, and division operator, as well as the sine, cosine, and square root. In addition to the input variables, the set of terminal nodes also includes numerical constants generated from the interval $[-1, 1]$. Two stopping criteria were used, the maximum number of evaluations and the maximum number of evaluations without improvement in the best individual. The maximum number of evaluations was set to 500 000, whereas the number of evaluations without improvement was set to 25 000. The fitness function being minimized is the mean squared error between the expected outputs and the output values obtained by the evolved expression for all input values. Each experiment was executed 50 times to obtain statistically significant results.

## 5 EXPERIMENTAL RESULTS

### 5.1 RESULTS FOR SYNTHETIC BENCHMARKS

In the first set of experiments, GP is tasked with finding the expression for ten synthetic benchmark functions, using the three specified maximum tree depths (6, 8, and 10). In every experiment, the fitness of original expressions is denoted as *original*, whereas the fitness of simplified expressions is denoted as *final*. Three scenarios were conducted in the experiments. The first scenario, where GP was executed with a random initial population to find the symbolic expression for the given data, is denoted as *Random*. The expressions obtained from this run are then used as the seeds for the initial population of a second GP execution denoted as *SubtreeInit*. Since the comparison between those two approaches could be considered unfair, we include an additional experiment in which GP was given twice the number of evaluations than the *Random* variant to compensate for the additional time that is used when rerunning the GP with a new initial population. This scenario is denoted as *Random2x*.

Table 1 gives the results for the test set. The experiments demonstrate that the obtained results depend quite heavily on the maximum tree depth. Indeed, for expressions of depth 6, the results demonstrate that the initialization with the winning trees did not improve the results in most cases. However, as the depth of the expressions increases, the results of the *SubtreeInit* approach improve. This is most evident when using a tree depth of size 10. For the tree depth of 10, the results obtained

by *SubtreeInit* are equally good or better than the results obtained by both *Random* or *Random2x* for all the tested functions. Another interesting result is that the method initialized with winning trees outperformed GP with the random initialization, which was given twice the number of evaluations. This shows that the improvement in the results is not a consequence of the extra time that GP was given. Rather, the good solutions obtained seem to be a direct consequence of the winning trees inserted in the starting population.

Such a behavior can be explained so that the winning trees obtained for tree depth six will be quite small and thus very general. Therefore, GP will still struggle to combine such subtrees in a meaningful way. However, for the larger tree depths, a larger portion of the solution will be transferred. Naturally, this subtree does not have a good fitness by itself since it represents only a part of the expression. However, it is more "specialized" for the considered problem.

Table 1: Test set results.

| depth | | 6 | | 8 | | 10 | |
|---|---|---|---|---|---|---|---|
| | | original | final | original | final | original | final |
| **Keijzer 4** | Random | 0.364598 | 0.389749 | 0.323364 | 0.349782 | 0.34916 | 0.389907 |
| | Random2x | 0.369 | 0.391 | 0.323 | 0.351 | 0.303 | 0.337 |
| | SubtreeInit | 0.404064 | 0.404064 | 0.296285 | 0.314635 | 0.307674 | 0.330624 |
| **Keijzer 12** | Random | 0.6998565 | 0.700004 | 0.6735125 | 0.700004 | 0.6719495 | 0.700004 |
| | Random2x | 0.697 | 0.700 | 0.674 | 0.700 | 0.659 | 0.700 |
| | SubtreeInit | 0.700004 | 0.700004 | 0.700004 | 0.700004 | 0.700004 | 0.700004 |
| **Keijzer 16** | Random | 2.59087 | 2.86854 | 1.336335 | 1.51365 | 1.2123 | 1.42172 |
| | Random2x | 2.840 | 3.082 | 1.232 | 1.420 | 1.158 | 1.230 |
| | SubtreeInit | 3.74226 | 3.99701 | 0.987748 | 1.08079 | 0.894961 | 0.982344 |
| **Korns 4** | Random | 0.020706 | 0.020706 | 0.009821 | 0.009821 | 0.008658 | 0.00921 |
| | Random2x | 0.013 | 0.013 | 0.012 | 0.012 | 0.011 | 0.011 |
| | SubtreeInit | 0.040262 | 0.040262 | 0.031405 | 0.031405 | 0.011096 | 0.011096 |
| **Korns 12** | Random | 1.01063 | 1.01063 | 1.00646 | 1.01892 | 0.986981 | 1.01063 |
| | Random2x | 1.010 | 1.098 | 0.995 | 1.011 | 0.991 | 1.011 |
| | SubtreeInit | 1.01063 | 1.01063 | 1.008765 | 1.01063 | 0.998089 | 1.01063 |
| **Nguyen 1** | Random | 9.72E-05 | 9.77E-05 | 0.082576 | 0.082576 | 0.135774 | 0.137373 |
| | Random2x | 0.000 | 0.000 | 0.018 | 0.018 | 0.103 | 0.110 |
| | SubtreeInit | 4.97E-15 | 5.74E-15 | 4.97E-15 | 5.74E-15 | 4.97E-15 | 5.74E-15 |
| **Nguyen 4** | Random | 59.94085 | 59.94105 | 63.46775 | 69.0745 | 48.5 | 51.867 |
| | Random2x | 15.998 | 15.998 | 57.950 | 65.081 | 28.058 | 30.807 |
| | SubtreeInit | 3.20736 | 3.24277 | 1.079874 | 1.079875 | 2.27039 | 2.298855 |
| **Nguyen 5** | Random | 0.196108 | 0.196108 | 0.209633 | 0.213252 | 0.167364 | 0.195794 |
| | Random2x | 0.122 | 0.122 | 0.075 | 0.081 | 0.226 | 0.237 |
| | SubtreeInit | 0.359085 | 0.395686 | 0.240072 | 0.244396 | 7.02E-17 | 7.02E-17 |
| **Nguyen 6** | Random | 0.294081 | 0.294081 | 0.371099 | 0.403021 | 0.353553 | 0.40881 |
| | Random2x | 0.436 | 0.454 | 0.258 | 0.276 | 0.379 | 0.426 |
| | SubtreeInit | 0.425501 | 0.442081 | 0.481001 | 0.524147 | 0 | 5.72E-16 |
| **Nguyen 12** | Random | 3.678065 | 3.892595 | 3.65844 | 4.038375 | 3.89702 | 4.29179 |
| | Random2x | 4.385 | 4.398 | 4.367 | 4.881 | 5.006 | 4.892 |
| | SubtreeInit | 5.40213 | 5.40213 | 4.907745 | 5.39591 | 0.712049 | 0.706132 |

UNIVERSALITY OF WINNING TREES

After the initial results, we can investigate whether the winning trees for one problem are universal, i.e., can they also be used to initialize the population when applying GP for a different problem. To test this hypothesis, we selected five function pairs and used the obtained individuals for one function to initialize the population of GP when optimizing the second function. The functions were selected so that in certain cases, similar functions are paired together (Nguyen 12 and Keijzer 16) while, in other cases, functions with completely different behavior are paired together (Nguyen 4 and Keijzer 4). Table 2 shows the results for this experiment. The first function represents the optimized function, while the second represents the one from which the subtrees were used for initialization (e.g., Keijzerf12_subtree_Kornsf12 denotes that we optimize function Keijzer 12, and we use subtrees from Korns 12).

The results generally suggest that the winning trees are not universal, since in most cases, the results obtained by the unmatched initialization procedure were worse than those obtained by either the ran-

Table 2: Test set results with the wrong subtree initialization.

| | depth | 6 | | | 8 | | | 10 | | |
|---|---|---|---|---|---|---|---|---|---|---|
| | | min | med | max | min | med | max | min | med | max |
| Keijzerf12_subtree_Kornsf12 | original | 0.572146 | 0.700004 | 0.701603 | 0.000349386 | 0.700004 | 1.42152 | 0 | 0.700004 | 3.10611 |
| | final | 0.572146 | 0.700004 | 0.700004 | 0.000349386 | 0.700004 | 0.701212 | 0 | 0.700004 | 0.705602 |
| Keijzerf16_subtree_Nguyenf12 | original | 1.13344 | 2.684715 | 15.9416 | 0.448655 | 1.10806 | 5.37345 | 0.407764 | 0.99726 | 5.01995 |
| | final | 1.35741 | 2.844835 | 16.0473 | 0.561838 | 1.21787 | 3.76721 | 0.478139 | 1.127875 | 5.58704 |
| Nguyenf4_subtree_Keijzerf4 | original | 5.99974E-13 | 1.957375 | 118.984 | 4.31563E-13 | 7.54E-13 | 45.4089 | 4.53E-13 | 0.553775 | 41.3374 |
| | final | 4.69576E-13 | 1.957375 | 123.222 | 4.49482E-13 | 7.3E-13 | 51.4943 | 4.53E-13 | 0.589304 | 41.3374 |
| Nguyenf5_subtree_Nguyenf1 | original | 7.02167E-17 | 0.348687 | 0.428466 | 7.02167E-17 | 0.348284 | 0.428484 | 7.02E-17 | 0.20722 | 0.428465 |
| | final | 7.02167E-17 | 0.354029 | 0.440514 | 7.02167E-17 | 0.394857 | 0.440291 | 7.02E-17 | 0.23288 | 0.428465 |
| Nguyenf6_subtree_Kornsf4 | original | 0 | 0.5507845 | 0.647055 | 0 | 0.502387 | 0.647055 | 0 | 0.414604 | 0.647055 |
| | final | 0 | 0.5817115 | 0.655615 | 0 | 0.551956 | 0.647055 | 0 | 0.465653 | 0.647055 |

Table 3: Test set results with random initialization, real-world datasets.

| | depth | 6 | | | 8 | | | 10 | | |
|---|---|---|---|---|---|---|---|---|---|---|
| function | | min | med | max | min | med | max | min | med | max |
| airfoil | original | 5.396 | 6.934 | 14.975 | 4.538 | 5.966 | 15.640 | 3.948 | 5.783 | 77.790 |
| | final | 5.455 | 7.487 | 17.482 | 5.223 | 6.150 | 16.583 | 4.799 | 5.923 | 8.507 |
| ccpp | original | 4.771 | 7.433 | 33.544 | 4.524 | 6.125 | 16.189 | 4.126 | 5.194 | 45.341 |
| | final | 5.059 | 7.668 | 35.548 | 4.827 | 6.710 | 16.824 | 4.741 | 5.946 | 51.875 |
| concrete | original | 8.024 | 11.972 | 19.390 | 9.887 | 12.109 | 175.524 | 10.123 | 13.027 | 20.201 |
| | final | 7.856 | 12.006 | 18.627 | 9.185 | 12.462 | 19.769 | 9.304 | 12.594 | 22.162 |
| wine_red | original | 0.637 | 0.678 | 0.733 | 0.639 | 0.675 | 0.966 | 0.653 | 0.684 | 0.835 |
| | final | 0.647 | 0.727 | 0.850 | 0.653 | 0.722 | 0.834 | 0.690 | 0.739 | 0.847 |
| wine_white | original | 0.684 | 0.720 | 0.994 | 0.688 | 0.715 | 0.809 | 0.667 | 0.709 | 0.793 |
| | final | 0.702 | 0.775 | 0.883 | 0.711 | 0.776 | 0.895 | 0.720 | 0.760 | 0.883 |

dom or subtree initialization procedures. Therefore, winning trees seem exclusive to the considered problem and cannot be reused across different problems. As such, this is an indication that winning trees represent a specialization of building blocks in GP.

## 5.2 RESULTS FOR THE REAL-WORLD DATASETS

Table 3 gives the results obtained by GP when using random initialization, whereas Table 4 presents the results obtained when the population is initialized by winning trees obtained from the initial run. For the first two test problems, namely AFN and CCP, it is evident that initializing the population with winning trees improves the overall results. For the other three, the results are more evenly matched. This is especially true for the WIR and WIW problems, where both methods achieved an almost equal performance.

Table 4: Test set results with subtree initialization, real-world datasets.

| | depth | 6 | | | 8 | | | 10 | | |
|---|---|---|---|---|---|---|---|---|---|---|
| function | | min | med | max | min | med | max | min | med | max |
| airfoil | original | 5.182 | 5.795 | 138.921 | 4.483 | 5.495 | 8.952 | 4.405 | 5.407 | 64.376 |
| | final | 5.536 | 5.994 | 139.233 | 4.964 | 5.729 | 7.469 | 4.557 | 5.670 | 64.429 |
| ccpp | original | 4.635 | 5.561 | 15.990 | 4.411 | 5.330 | 17.331 | 4.137 | 4.707 | 15.980 |
| | final | 4.637 | 6.058 | 15.990 | 4.972 | 5.967 | 17.331 | 4.701 | 5.390 | 15.980 |
| concrete | original | 9.489 | 13.025 | 20.428 | 9.535 | 11.953 | 19.549 | 8.329 | 12.934 | 96.710 |
| | final | 9.474 | 12.145 | 19.095 | 8.735 | 11.856 | 19.152 | 7.819 | 12.634 | 25.108 |
| wine_red | original | 0.644 | 0.675 | 0.778 | 0.645 | 0.671 | 0.698 | 0.632 | 0.676 | 0.882 |
| | final | 0.662 | 0.717 | 0.832 | 0.652 | 0.727 | 0.834 | 0.652 | 0.717 | 0.825 |
| wine_white | original | 0.695 | 0.728 | 0.857 | 0.688 | 0.712 | 2.780 | 0.687 | 0.705 | 20.870 |
| | final | 0.720 | 0.770 | 0.951 | 0.726 | 0.771 | 0.892 | 0.712 | 0.769 | 0.904 |

## 5.3 EXPRESSION COMPLEXITY ANALYSIS

Besides the test performance, another perspective worth considering is their complexity, i.e., the sizes of the obtained expressions. Table 5 shows the median of expression sizes for all synthetic test functions. First, it can be noticed that the applied simplification and pruning methods were effective in reducing the complexity of the expressions. On average, a reduction of 40% was achieved across all the experiments after both simplification methods were applied. Although for some functions, like Korns 12, the method simplifies the expression to only a single node, this result is not indicative of simplification only. The reason is that GP was, in this case, actually unable to fit this function well, and the expression it obtained does not perform better than only a single constant node that is obtained after simplification.

The second important observation is that the expressions that are evolved when initializing the population with winning trees are, in most cases, significantly smaller than when using random initialization. On average, when considering all the experiments, initialization with winning trees reduces the size of the evolved expressions by around 50%. This demonstrates that a good initialization leads to a smaller occurrence of bloat. Still, it should be noted that it does not eliminate bloat completely since the obtained solutions could be simplified further.

Table 5: Synthetic problems, solution sizes with random init, random init with twice the evaluations, and subtree init, size median. The first occurrence of the smallest solution is denoted in bold style.

| depth | | Random | | | Random2x | | | Subtree | | |
|---|---|---|---|---|---|---|---|---|---|---|
| | | 6 | 8 | 10 | 6 | 8 | 10 | 6 | 8 | 10 |
| Keijzer 4 | original | 29.5 | 62 | 123 | 28 | 69.5 | 135.5 | **8** | 52 | 67 |
| | simplify | 27 | 54.5 | 109 | 25.5 | 64 | 122 | 8 | 46 | 63 |
| | prune | 22.5 | 41.5 | 62 | 21 | 43 | 87 | 8 | 34 | 42 |
| | simplify | **21** | **39** | **54** | **19** | **38.5** | **81** | 8 | **33** | **36** |
| Keijzer 12 | original | 17.5 | 32 | 51.5 | 17 | 36.5 | 52 | **3** | **3** | **3** |
| | simplify | 15.5 | 30 | 50 | 15 | 33 | 50.5 | 3 | 3 | 3 |
| | prune | 5 | 9 | 5 | 5 | 7 | 8 | 3 | 3 | 3 |
| | simplify | **3** | **7** | **3** | **3** | **3** | **3** | 3 | 3 | 3 |
| Keijzer 16 | original | 53 | 106 | 142.5 | 59.5 | 110 | 153.5 | 53 | 66 | 158.25 |
| | simplify | 51 | 99 | 134 | 55 | 108 | 144 | 51.5 | 62.25 | 147.5 |
| | prune | 47 | 92 | 121 | 54.5 | 103.5 | 130.5 | 48.5 | 59.25 | 140.75 |
| | simplify | **46** | **87** | **111.5** | **52.5** | **100.5** | **125.5** | **46.5** | **57.5** | **139.75** |
| Korns 4 | original | 22 | 28 | 43.5 | 23 | 32 | 42 | 11 | 14 | 33 |
| | simplify | 12 | 20 | 31.5 | 12 | 21.5 | 27 | 8.5 | **11** | 24 |
| | prune | **11.5** | 19.5 | 30.5 | 11.5 | 21.5 | 26 | 8.5 | 11 | 21.5 |
| | simplify | 11.5 | **19** | **29.5** | **11** | **21** | **23** | **8** | 11 | **20** |
| Korns 12 | original | 35 | 66.5 | 98.5 | 36 | 64 | 94 | 7 | 44 | 96.5 |
| | simplify | 25.5 | 57.5 | 90 | 31 | 56 | 87.5 | **1** | 36.5 | 83 |
| | prune | 7 | 9 | 8 | 8 | 11 | 8 | 1 | 7 | 8 |
| | simplify | **1** | **1** | **1** | **1** | **1** | **1** | 1 | **1** | **1** |
| Nguyen 1 | original | 20.5 | 32.5 | 72 | 24.5 | 41.5 | 55.5 | 11 | 11 | 11 |
| | simplify | **13** | **23** | 56.5 | 15 | 32.5 | 43 | **9** | **9** | **9** |
| | prune | 13 | 23 | 51.5 | **13** | **26** | 38 | 9 | 9 | 9 |
| | simplify | 13 | 23 | **50** | 13 | 26 | **36** | 9 | 9 | 9 |
| Nguyen 4 | original | 54.5 | 107.5 | 183 | 50 | 113 | 232 | 28 | 33 | 46.5 |
| | simplify | 49 | 95.5 | 165 | 47 | 104 | 217 | **23** | 27.5 | **39** |
| | prune | **47** | 91.5 | 154 | 45.5 | 99 | 193 | 23 | 27 | 39 |
| | simplify | 47 | **89.5** | **144** | **44.5** | **98** | **188** | 23 | 26.5 | 39 |
| Nguyen 5 | original | 25 | 46.5 | 99 | 24 | 40.5 | 78 | 10.5 | 19 | 11 |
| | simplify | 21.5 | 37.5 | 86.5 | 20 | 32 | 72 | 9 | 16.5 | **9** |
| | prune | 17.5 | 23 | 61.5 | 14 | **28** | 46 | **7.5** | 14.5 | 9 |
| | simplify | **17** | **16.5** | **59.5** | **13** | 28 | **37.5** | 7.5 | **12** | 9 |
| Nguyen 6 | original | 17.5 | 31 | 58 | 19 | 31 | 54 | 11 | 17 | **9** |
| | simplify | 14 | 29.5 | 55 | 17.5 | 27 | 49.5 | 10.5 | 16.5 | 9 |
| | prune | 11.5 | 25.5 | 36.5 | 16 | 19 | 33.5 | 9 | 11.5 | 9 |
| | simplify | **9.5** | **22.5** | **31.5** | **15** | **17** | **29.5** | **9** | **10.5** | 9 |
| Nguyen 12 | original | 37.5 | 77 | 105.5 | 43 | 89 | 111.5 | 12 | 23 | 56 |
| | simplify | 33 | 72.5 | 97 | 39.5 | 79.5 | 103.5 | **11** | 20 | 51.5 |
| | prune | 31 | 66.5 | 86 | 38.5 | 72 | 94 | 11 | 19 | 42.5 |
| | simplify | **30** | **65** | **81.5** | **38** | **70** | **92** | 11 | **18** | **38.5** |

Table 6 shows the summary of the obtained expression sizes for real-world problems after the execution of the simplification and pruning methods. The expressions obtained using the winning trees for initialization did not always lead to smaller expressions for these problems. This happens for the WIW and WIR datasets, in which the expressions obtained with the initialization by winning trees were larger than those obtained with random initialization. A probable cause for this is that the algorithm quickly converged to a good region of solutions and simply started to bloat and overfit on the training set. On the WIW dataset, we observe some spikes on the test set for two experiments (random initialization with the tree depth of 6 and subtree initialization with the three depth of 8). At this point, it seems that the method started to overfit, but nevertheless, in the next iterations, solutions that performed well on the test set were evolved. For CST, the expressions are of similar size, whereas initialization with winning trees resulted in smaller expressions being evolved for the other two problems. In those cases, the algorithm converged more slowly, and as such, it seems that they did not have the chance to overfit the training set.

Table 6: Real-world datasets, solution sizes with random and subtree initialization, size median. The first occurrence of the smallest solution is denoted in bold style.

| depth | | **Random** | | | **Subtree** | | |
| --- | --- | --- | --- | --- | --- | --- | --- |
| | | 6 | 8 | 10 | 6 | 8 | 10 |
| **Airfoil** | original | 69 | 154 | 233.5 | 53.5 | 77.5 | 201 |
| | simplify | 67 | 143.5 | 230 | 49.5 | 73.5 | 191.5 |
| | prune | 54 | 114 | 169 | 34.5 | 44.5 | 141 |
| | simplify | **48** | **105.5** | **154.5** | **28** | **37.5** | **134.5** |
| **Ccpp** | original | 45.5 | 107 | 180.5 | 39.5 | 72.5 | 140 |
| | simplify | 44.5 | 104.5 | 171.5 | 38.5 | 72 | 138 |
| | prune | 39 | 77 | 101 | 33 | 58 | 66 |
| | simplify | **36** | **72.5** | **86** | **29.5** | **51** | **55.5** |
| **Concrete** | original | 43.5 | 80.5 | 157.5 | 40.5 | 77 | 156.5 |
| | simplify | 43.5 | 79.5 | 152.5 | 40.5 | 76.5 | 154 |
| | prune | 32 | 52.5 | 77 | 29 | 48.5 | 79 |
| | simplify | **28** | **45.5** | **64** | **24** | **42** | **72** |
| **Red wine** | original | 29 | 51 | 64 | 20 | 45 | 83 |
| | simplify | 29 | 50.5 | 64 | 20 | 44.5 | 82 |
| | prune | 17 | 21.5 | 19 | 14 | 19 | 20 |
| | simplify | **13** | **17.5** | **14** | **11.5** | **13.5** | **12** |
| **White wine** | original | 31 | 50 | 84.5 | 18.5 | 42 | 96 |
| | simplify | 31 | 49.5 | 83 | 18.5 | 41.5 | 95.5 |
| | prune | 16 | 20 | 22.5 | 12 | 15 | 18.5 |
| | simplify | **11.5** | **13.5** | **14** | **9** | **9** | **10** |

## 6    CONCLUSIONS AND FUTURE WORK

This paper analyses the influence of simplification in GP for regression problems. First, we investigate how various simplification procedures can result in smaller, yet fit solutions. Afterward, we use the parts of the obtained solutions (denoted as winning trees) to seed the population for new runs. Our results indicate that winning trees can help reach better solutions in a smaller number of evaluations. This approach works especially well for cases where solutions have larger depth as then, winning trees have sufficient "space" to be properly adapted, and where the tree depth size makes it more challenging for the GP process to evolve highly fit solutions.

As this paper proposes a new concept – winning trees, there are multiple possible future research directions. It is possible to look at winning trees as a building block specialization with constraints on size and fitness. Still, one could question our design choices in several places. For instance, why considering the problems we consider, why not tune more or the GP's parameters, why go for subtrees (winning trees) of depth half the maximum tree depth. Furthermore, the insertion and application of subtrees in the new population can also be performed to provide more influence to the evolutionary process. We consider the investigation of those alternatives a natural next step.

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

Table 7: Mapping rules for simplification.

| | | | |
|---|---|---|---|
| $*X1 \to X$ | $-XX \to 0$ | $+X*YX \to *+1YX$ | $+*XY*XZ \to *X+YZ$ |
| $*1X \to X$ | $-X0 \to X$ | $+X*XY \to *+1YX$ | $+*YX*ZX \to *X+YZ$ |
| $*X0 \to 0$ | $/X1 \to X$ | $+*YXX \to *+1YX$ | $+*\sin A \cos B * \cos A \sin B \to \sin +AB$ |
| $*0X \to 0$ | $/*XYY \to X$ | $+*XYX \to *+1YX$ | $-*\sin A \cos B * \cos A \sin B \to \sin -AB$ |
| $+0X \to X$ | $*/XYY \to X$ | $-X--1Y1 \to +XY$ | $-*\cos A \cos B * \sin A \sin B \to \cos +AB$ |
| $+X0 \to X$ | $-X-0Y \to +XY$ | $\sin 0 \to 0$ | $+*\cos A \cos B * \sin A \sin B \to \cos -AB$ |
| $/XX \to 1$ | $+X--1Y1 \to -XY$ | $\sin -0X \to -0 \sin X$ | $*D_2 * \sin A \cos A \to \sin *2A$ |
| $\cos 0 \to 1$ | | | |

Table 8: Synthetic benchmark functions.

| Name | Definition |
|---|---|
| Keijzer 4 | $f(x) = 0.3 * x * \sin(2 * \pi * x)$ |
| Keijzer 12 | $f(x, y) = xy + \sin((x - 1)(y - 1))$ |
| Keijzer 16 | $f(x, y) = \frac{x^3}{5} + \frac{y^3}{2} - y - x$ |
| Korns 4 | $f(x) = -2.3 + (0.13 * \sin(x))$ |
| Korns 12 | $f(x, y) = 2.0 - (2.1 * \cos(9.8 * x) * \sin(1.3 * y))$ |
| Nguyen 1 | $f(x) = x^3 + x^2 + x$ |
| Nguyen 4 | $f(x) = x^6 + x^5 + x^4 + x^3 + x^2 + x$ |
| Nguyen 5 | $f(x) = \sin(x^2)\cos(x) - 1$ |
| Nguyen 6 | $f(x) = \sin(x) + \sin(x + x^2)$ |
| Nguyen 12 | $f(x, y) = x^4 - x^3 + \frac{y^2}{2} - y$ |

## A  ADDITIONAL RESULTS

The mapping rules for simplification in our experiments are given in Table 7, defined using the prefix notation. The left side of each line denotes the original pattern that the Compare-Match algorithm searches for in the expression. On the other hand, the right side represents the expression by which the pattern in the original expression will be replaced.

---

**Algorithm 1** Compare-Match procedure

---

**Input:** P – pattern (left side of the rule), T – expression
**Output:** Boolean: $true$ if the rule has changed, $false$ otherwise.

**Function** Compare
**if** root(P) = $null$ —— root(T) = $null$ **then**
    **return** $true$
**end if**
**if** !(Map(P, T)) **then**
    **return** $false$
**end if**
**for** $(i := 0; i < degree(P); i++)$ **do**
    **if** !Compare(P-¿child[i], T-¿child[i]) **then**
        **return** $false$
    **end if**
**end for**
**return** $true$

**Function** Match
**if** Compare(P, T) **then**
    Replace pattern P in T with the simplified expression
    **return** $true$
**end if**
Boolean matched := $false$
**if** T != $null$ **then**
    **for** $(i := 0; i < degree(T); i++)$ **do**
        matched := Match(P, T-¿child[i]) —— matched
    **end for**
**end if**
**return** matched

---

CONVERGENCE

Figure 1 shows the GP convergence results for real-world problems. The plots represent the median of the best individuals from the 50 performed runs. The convergence is shown both on the training

Table 9: Real-world datasets.

| Abbr. | Dataset | # of instances | # of features |
|---|---|---|---|
| AFN | Airfoil self-noise | 1 503 | 6 |
| CCP | Combined cycle power plant | 9 568 | 4 |
| CST | Concrete strength | 1 030 | 9 |
| WIR | Wine quality, red wine | 1 599 | 12 |
| WIW | Wine quality, white wine | 4 898 | 12 |

set and on the test set, where in each iteration, the best individual from the training set was selected and evaluated on the test set.

The problems can be roughly grouped into two categories by their behavior. For the first three problems, we observed that some of the methods still did not converge in the given time and were improving the solutions. For the final two problems, namely WIR and WIW, the algorithms converged quite quickly, and for most of the run, the solutions did not improve significantly. The best convergence was achieved on the training set by GP initialized with winning trees and with the maximum tree depth of 10. The convergence also improves when using winning trees for the other tree depths, although not as consistently. On the test set, it is evident that these methods reach good solutions faster than random initialization. This means that inserting existing genetic material into the initial population did not lead to overfitting but rather helped speed up the convergence.

### STABILITY AND DIVERSITY OF THE SYNTHETIC SOLUTIONS

Figure 2 shows the boxplot results for the synthetic benchmark datasets. This figure aims to determine the stability and dissipation of the results between the tested methods. For the smallest depth of 6, the initialization with winning trees did not improve the stability of the results. In general, the results obtained by this method were, in several cases, even more dispersed than with random initialization (for example, in cases of the Korns 4 and Keijzer 16 functions). In cases where the results had a small dissipation, they were generally worse than the results obtained by random initialization. For tree depth of 8, the results tend to improve in some cases, but still, the initialization with winging trees does not demonstrate a clear dominance over the GP results obtained by random initialization. However, the situation largely improves with the tree depth of 10. In this case, the initialization with winning trees leads to significantly less dispersed results than all the other experiments. This is especially evident in the Nguyen type functions, in which the dispersion is almost negligible compared to the other experiments. This suggests that for the depths of 6 and 8, the extracted winning trees were too small to provide useful information that could more efficiently guide GP towards better solutions.

To evaluate the diversity of solutions, we can compare the obtained solutions semantically (i.e., if the resulting trees "look" the same) and functionally (if the trees give the same fitness value as output). For ten synthetic benchmarks, for 21 out of 30 cases, initialization with winning tress gives less semantically diverse solutions. Additionally, for 21 out of 30 cases, initialization with winning trees gives less functionally diverse solutions.

### STABILITY AND DIVERSITY OF THE REAL-WORLD SOLUTIONS

Figure 3 shows the boxplot results for the real-world problems. These plots suggest that for these problems, the initialization with winning trees again provides several benefits. However, this again becomes evident only for the two larger tree sizes, as for the tree depth of 6 initialization with winning trees did not always perform better. The initialization with winning trees resulted in less dispersed and more stable results for the other two tree depths, confirming the results obtained for the synthetic benchmarks.

Finally, for 4 out of 15 cases, initialization with winning tress gives less semantically diverse solutions. For 6 out of 15 cases, initialization with winning trees gives less functionally diverse solutions.

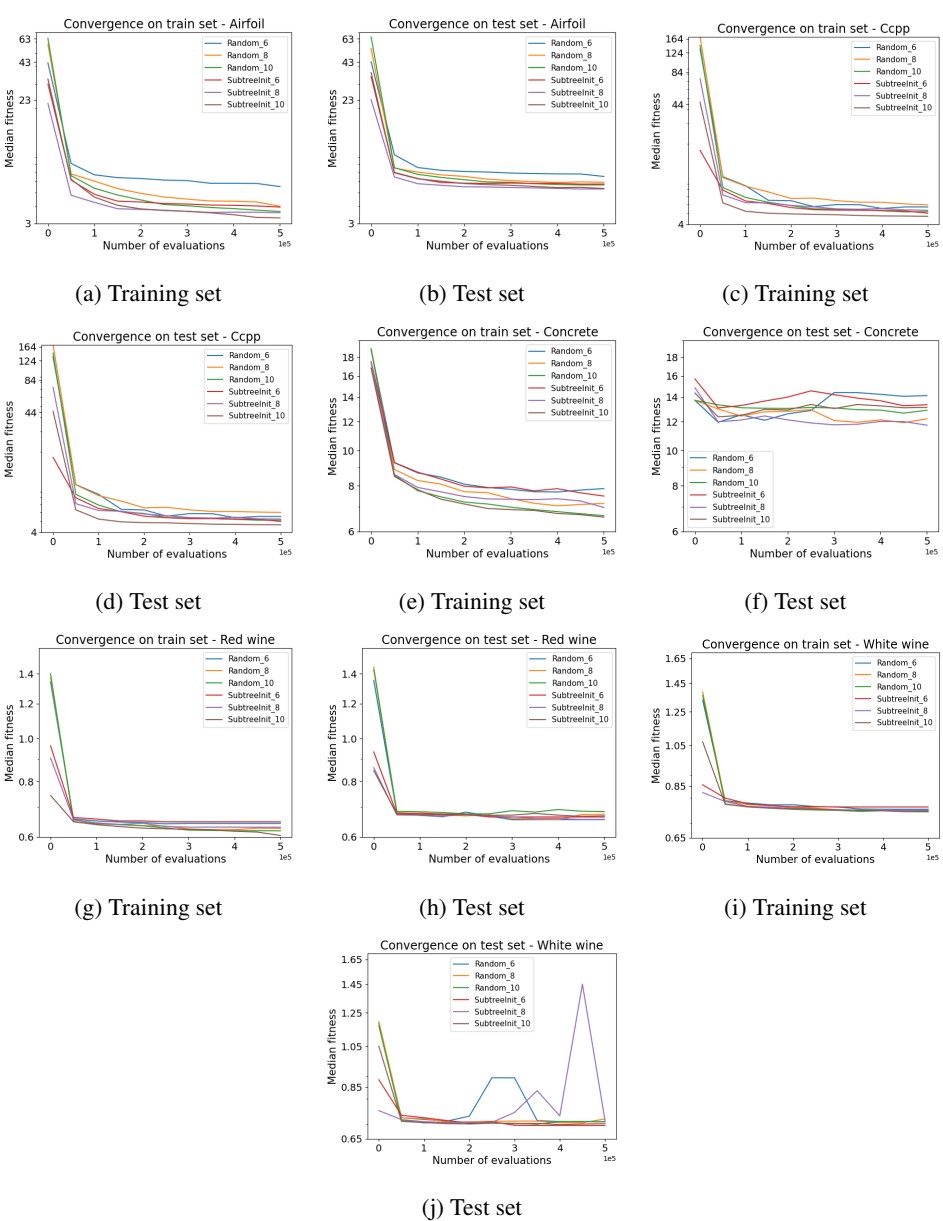

Figure 1: Algorithm convergence comparison for the real-world problems.

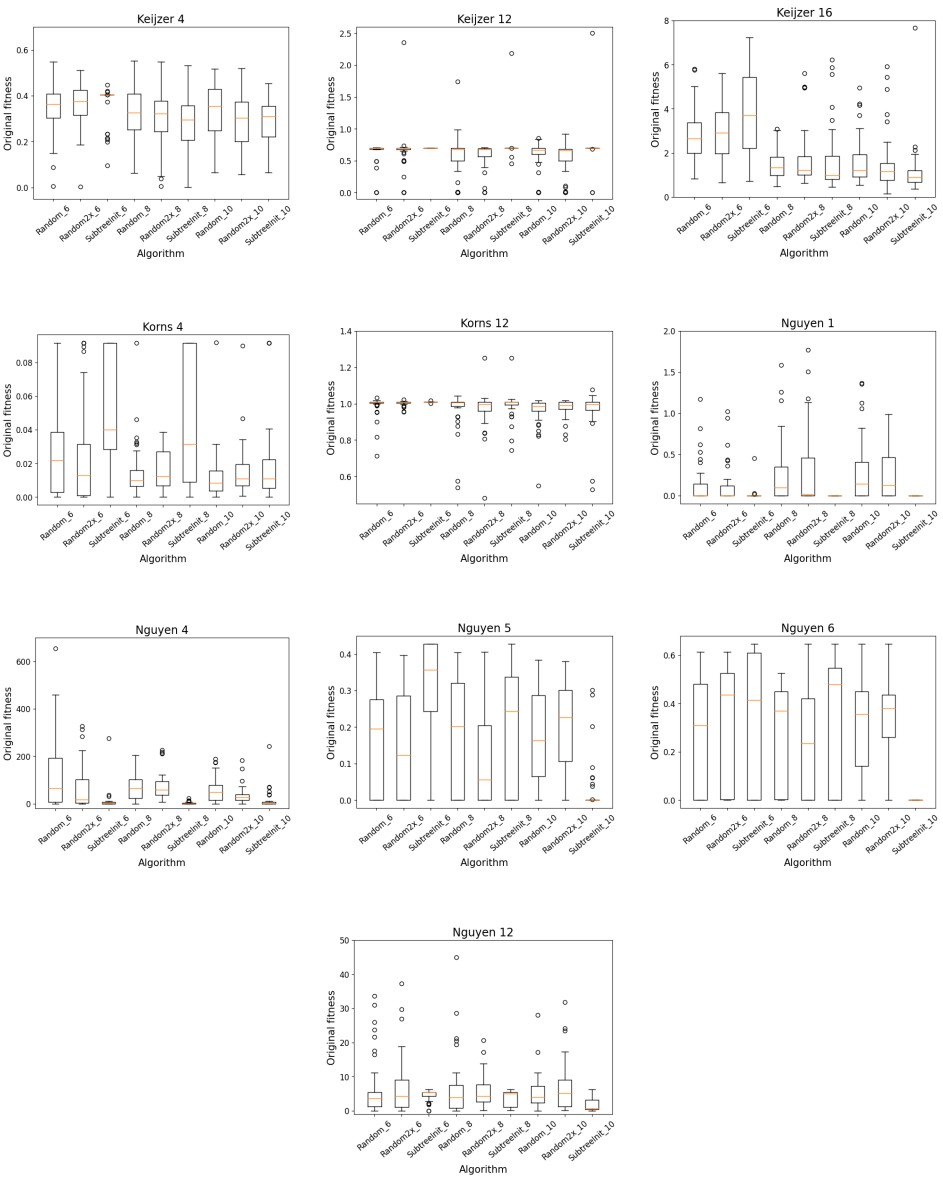

Figure 2: Synthetic problems, boxplot results.

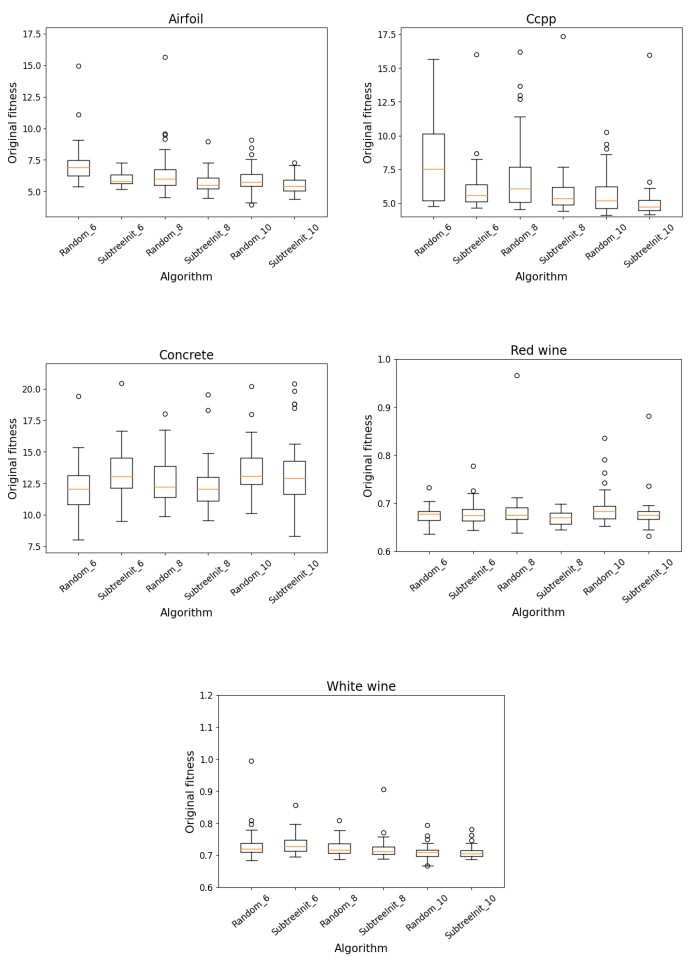

Figure 3: Real-world problems, boxplot results.

