# OpenReview forum: "Building the Building Blocks: From Simplification to Winning Trees in Genetic Programming"
_ICLR.cc/2022/Conference — ICLR 2022 Submitted_

### Official Review · Reviewer_77nm · 2021-10-27

**Correctness:** 2
**Technical Novelty And Significance:** 2
**Empirical Novelty And Significance:** 2
**Recommendation:** 3
**Confidence:** 5

**Main Review:**

The paper presents good writing style and is easy to follow. It devotes however the whole 9 pages to written explanations and a few numeric tables of results, relegating fundamental information such as part of the algorithm and the entire results plots to the appendix. That is right, there is not a single figure in the paper. This is exacerbated by the experimental results being presented as 6 (six!) purely numerical tables accounting for roughly 3 full pages, with minimal discussion and almost absent captioning, which makes the experimental evaluation unnecessarily hard to assess. The contributions presented are themselves limited, which does not support this choice of presentation. Before I can vouch for its publication, I would require the authors to present the results in a more readable way which does not rely on Appendix material for interpretation or reproducibility.
From a more technical perspective, the paper offers limited support for its arguments. Its main contributions are (i) a rule-based tree simplification method including both algebraic and pruning techniques, and (ii) an archive-based restart strategy.
However, neither contribution is compared in the experimental results against the current state of the art, but only against standard GP. This offers a limited insight on its relative advantage, but no clear perspective to the method's actual applicability and utility.

Main questions that need to be addressed by the authors (in the paper and with experimental results) prior to considering this work ready for publication:
- What is the practical, empirical difference between the literature method of simplification with removal and the proposed substitution with an operation-neutral constant? I.e., what is the difference between removing a sub-tree from a node with e.g. a multiplication operator, and substituting it with a constant-one (1) leaf?
- The pruning procedure accepts an arbitrary 15% degradation in fitness on the simplified tree, before looping on itself. In your experiments, is the 15% degradation a global budget over successive loops (i.e. the final simplified tree cannot be worse than 15% on the original tree) or a per-loop limit (which leads to a compounding effect as the solution can worsen by 15% per iteration)?
- Can you highlight and discuss the performance degradation as depicted above across the Results tables? How did you come up with 15%? Is this hyperparameter robust to small variations?
- Only aggregated final performance is presented, which limits our understanding of the learning process over time. Please include a standard plot of performance improvement over time for all methods.
- A budget of 500'000 individuals is significant, especially on a low-dimensional benchmark of limited complexity, and with a mutation rate of 30%. How do the standard GP and your contribution fare against random guessing? I would expect a baseline of generating 500'000 individuals, and include the performance of the best-so-far over time in the comparison.
- Figures 1 and 2 (Appendix) lack any discussion or interpretation. For example, what is happening with Nguyen 5 and 6 in Figure 2? Why the high variance for everyone but 0 for SubTreeInit? The paper lacks a proper discussion at this stage.
- How does this technique stand against standard restart strategies? What about niching, as each run could be considered as a niche for the final restarted run? They are entirely missing from the literature review. Moreover, the global consensus on evaluating restart strategies is to maintain a standard useful budget of individual evaluation constant for all method, with the restart being able to reset as many times as seen fit. Instead the current experiments only restart once, and for "fairness" doubles the available fitness evaluations from 500'000 to 1'000'000 evaluations. What if the standard GP reaches convergence by 400'000? How would this be fair?
- Building on that, it is clearly stated (Section 4.2) that the limit for convergence is 25'000 evaluations without improvement: assuming this means early termination if convergence is detected (as usual), what is the budget actually used by GP vs. the proposed contribution over the 50 runs?

Minor points:
- It is part of the authors' responsibilities to recognize which results are most compelling for their arguments and present those in a more digestible form. The corresponding plots from the Appendix should be moved to the main text. The full results tables could instead be moved to the Appendix.
- In Section 5.2, I would relax the claim of "it is evident that [...] improves the overall results" from a qualitative statement to a quantitative statement (measured performance increase).
- In Section 5.3 you mention "a single constant node that is obtained after simplification". Section 3.3 however states the minimal heights for trees being 2 after simplification. I believe this may be a misunderstanding, but it shows how the exposition is perceived as unnecessarily complicate, and a requirement for all data required for reproducibility to be found in a single place in a clear format (which I propose being another requirement for acceptance).

**Summary Of The Paper:**

This paper presents two optimizations to standard GP: a tree pruning with introduction of operation-neutral nodes, and a restart strategy reusing a population of best trees from past runs. It presents results on a broad set of very simple regression problems.

**Summary Of The Review:**

Let me state first that I fully support further research in evolutionary computation and especially genetic programming, which I believe could develop in a feasible alternative (even extension) to neural networks in real-world scenarios where access to labeled data is insufficient. This paper proposes an optimization to GP that could improve its performance, which is its main limiting factor to a broader application. Nonetheless I feel the present contributions to be (i) relatively minor (just simplification and restart), (ii) not particularly novel (again, just simplification and restart), and (iii) insufficiently supported by the experimental section (no baselines, no state of the art comparison, no clear exposition of the relevance of the datasets, extremely hard to read presentation). Overall the sensation is that of a work in progress, which may very possibly lead to a crucial contribution down the line, but is not yet ready for publication especially at a major conference.

---

### Official Review · Reviewer_mNLH · 2021-10-29

**Correctness:** 2
**Technical Novelty And Significance:** 2
**Empirical Novelty And Significance:** 2
**Recommendation:** 3
**Confidence:** 4

**Main Review:**

The concept of winning trees in GP seems interesting, but this paper still suffers from the issues of 1) insufficient/unclear presentation and 2) limited experiments. My concerns include:
1.	In section 3.2, the authors propose a novel pruning method for GP. But the effectiveness of this pruning method has not been verified via experiments. Especially, their approach allows pruning even if it reduces the fitness compared to the original expression. By doing so, although the obtained expression can be further simplified, the fitness of the expression might be dramatically deteriorated. In most cases, the solution accuracy, i.e., the fitness, is the main objective of method.
2.	In the pruning method, the degraded limit is set to be 15%. It seems that this value is very larger, which will make the fitness of solutions deteriorate quickly. Is there any adaptive method to set this value, e.g., a simulated annealing way?
3.	The winning tress should be strictly defined, rather than just giving its conceptual definition. Without a strict mathematical definition together with some theoretical analysis, it is difficult to fully understand this new concept. For example, it is unclear how many winning trees can be obtained by GP for a specific problem, and the existence of the winning tree can be guaranteed?
4.	It is also not very clear how to obtain a winning tree, and how a winning tree constrains on the fitness value and the tree size.
5.	The proposed approach seems like the building blocks with restart search strategy. The advantages of the proposed approach comparing to other previous ones should be further highlighted.
6.	The parameter settings in the experiments should be further clarified. Especially, it seems that the influence of winning trees on the performance of GP relies on the tree depths. Thus, it is better to give more experimental results by setting tree depths with more values, e.g., 12, 15, 20, etc.
7.	What kind of results in Tables are? Are they the fitness values (i.e., the mean squared errors) of the best solution?
8.	From the experimental results, e.g., those of Korns 12, it is hard to say winning trees can enable GP to find better expressions for regression problems, which significantly limit its practicability.
9.	Some minor revisions are needed, e.g., what’re WIW, WIR, WIW? In Algorithm 1, “P-?child” should be “P->child”.

**Summary Of The Paper:**

This paper discusses the influence of simplification and pruning in GP for regression problems. A novel pruning method is presented, and a new concept termed as winning trees is proposed. Experiments are conducted based on ten one or two-dimensional synthetic symbolic regression problems and five real-world regression problems. Comparative results show that winning trees can help GP find better solutions in a smaller number of evaluations.

**Summary Of The Review:**

Although the idea seems interesting, this paper is not technically sound. Experimental results cannot fully support the main conclusion of this paper. Without a strict mathematical definition together with some theoretical analysis, it is difficult to fully understand the new concept of “winning trees” in GP.

---

### Official Review · Reviewer_Q7WW · 2021-10-30

**Correctness:** 1
**Technical Novelty And Significance:** 1
**Empirical Novelty And Significance:** 1
**Recommendation:** 1
**Confidence:** 5

**Details Of Ethics Concerns:**

The issues regarding ethics that I could find in this draft.

**Main Review:**

The paper presents several deficiencies in many aspects  -scientific, experimental, presentation- that render it unsuitable for publication. I will mention some of them that, in my opinion, are the most critical:

- Main hypothesis. Authors attempt to draw an analogy between GP and DL frameworks, in regard to the LTH[1], such that they claim that a similar phenomenon might exists as well in GP. However, authors have a misconception of what the LTH is really about: The LTH is an attempt of DL researchers at explaining why gradient optimization methods work so well at tuning deep networks parameters. It is not that the solution is already embedded in the deep network, as authors of this paper seem to put forward, but rather that such dense networks create pathways for a gradient to move around harsh search landscapes, escaping local minima, and ultimately reach very good solutions.

- Setting, related work and background. Authors set their contribution mostly around techniques aimed at tackling _bloat_ in GP trees, i.e. the phenomenon where GP trees being to grow in size, taking heavy tolls in evaluation costs, with minimal or null performance increase; and while it is true that authors present some pruning techniques, their method is also equally related to _restarting_ techniques and _multilayer_ GP models, where GP runs are executed serially with some modifications to initial populations. Multilayer GP is specially common in the area of feature extraction through GP; I suggest authors to review some works in the area [2,3], to attempt to draw a clear differentiation between their serialized GP model, and others already existing in the literature.

- Experimental assessment. Authors carried several experiments in order to test their approach, both in artificial, as well as real world, datasets. However, the results they present are difficult to read and/or incomplete. Table I, for example, it is not clear what exactly those values are... the average of the MSE of the 50 runs... ? presumably. There is no indication neither in table title, footnote, nor in the main text body, of what are those values, if higher or lower is better, etc. Even if it is MSE, those values alone do not provide any information regarding the performance of each method compared, because there is no standard deviation reported. The table is also difficult to read, because there is no indication (typically with a bold font) which one of the approaches performed better. Similar issues are present in all other result tables presented.


1. Frankle, J., & Carbin, M. (2018, September). The Lottery Ticket Hypothesis: Finding Sparse, Trainable Neural Networks. In International Conference on Learning Representations.
2. Lin, J. Y., Ke, H. R., Chien, B. C., & Yang, W. P. (2007). Designing a classifier by a layered multi-population genetic programming approach. Pattern Recognition, 40(8), 2211-2225.
3. Lin, J. Y., Ke, H. R., Chien, B. C., & Yang, W. P. (2008). Classifier design with feature selection and feature extraction using layered genetic programming. Expert Systems with Applications, 34(2), 1384-1393.

**Summary Of The Paper:**

On a technical level, the paper proposes two methods for Genetic Programming (GP) tree pruning, as well as a method for a population _restart_ that relies on both pruning methods. The aim of such approach is that in the second, or subsequent, GP runs, the GP algorithm starts from with an already-somewhat-good initial population of individuals that would allow GP to reach better solutions. On a scientific level, authors try to draw analogies between their approach, and recent advances in the deep learning (DL) community, regarding the Lottery Ticket Hypothesis (LTH). The authors complement their draft with a set of experimental evaluations on different standard datasets.

**Summary Of The Review:**

The work is insufficient in many aspects in order to be considered for acceptance.
- The main motivation behind the work seems unfounded due to a misunderstood concept from the deep learning area.
- Related work discussion is lacking in regard to some relevant works to the method presented.
- Experimental evidence is incomplete.

My recommendation is a clear reject-

---

### Official Review · Reviewer_W75E · 2021-11-02

**Correctness:** 3
**Technical Novelty And Significance:** 2
**Empirical Novelty And Significance:** 2
**Recommendation:** 3
**Confidence:** 3

**Main Review:**

Strengths
- The paper is easy to follow.
- The link with the lottery ticket hypothesis (LTH) is intriguing.
- The improvement of the proposed idea for depth $\geq 8$ is positive.

Weaknesses
- The paper doesn't seem to find the proposed winning trees very promising. Apart from the results in Table 1, the evidence seems to suggest the winning tree idea is not good?
  - In particular, although it's interesting to link to the LTH, I feel the link is very weak.
- Apart from the empirically study, the novelty on the method itself is limited: Only the pruning method is novel.
- Presentation can be improved.
  - All table titles should be more descriptive. E.g. You should describe what does the value in Table 1 mean.
- A paper would be good to cite: DreamCoder: Growing generalizable, interpretable knowledge with wake-sleep Bayesian program learning
  - It also has the idea of iterative learning of useful sub-expressions.

**Summary Of The Paper:**

Inspired by (Frankle & Carbin, 2019), the paper proposes to find winning trees from a set of GP solutions as initialisation for another GP procedure.
Winning tress are random sub-trees from GP solutions after simplification and a novel pruning strategy.
Results show that such procedure can lead to better regression performance when the expression depth is larger or equal to 8.
However, results also suggest that the discovered winning trees are specialised to specific tasks (i.e. fail to transfer) and cannot in general produce smaller trees.

**Summary Of The Review:**

I found the motivation of this paper is weak as its link to the LTH is not clear.
Further, results don't seem to suggest the idea of winning tree is very useful.
Therefore I suggest not accept this paper.
It would be interesting see if the author(s) can push this idea further (or implement in another manner) and show its usefulness - I'm looking forward to see more research on symbolic methods in the ML community.

---

### Author Response · Authors · 2021-11-22
**Response to reviewers**

We would like to thank the reviewers for their effort and comments. We agree with most of the comments and will continue research to improve our work.

---

### Decision · Program_Chairs · 2022-01-20

**Decision:**

Reject

**Comment:**

This paper explores the hypothesis that bloat can be prevented in Genetic Programming by identifying "winning subtrees" from simplified solutions, and use these to seed new GP runs. This idea is connected with the lottery ticket hypothesis in deep learning.

Reviewer are unanimous that the paper as it stands is not ready to publish. One big issue is that the empirical results are not particularly good. Another is that the conceptual foundations of the paper, in particular the parallell to the lottery ticket hypothesis, might be flawed. Nevertheless, there is much interesting research to do in this direction.